

# Enhancing credit card fraud detection with a stacking-based hybrid machine learning approach

Eyad Abdel Latif Marazqah Btoush[1], Xujuan Zhou[1], Raj Gururajan[1,2], Ka Ching Chan[1] and Omar Alsodi[1]

[1] School of Business, University of Southern Queensland, Brisbane, QLD, Australia
[2] School of Computing, Indian Institute of Technology, Kharagpur, Chennai, India

## ABSTRACT

The swift progression of technology has increased the complexity of cyber fraud, posing an escalating challenge for the banking sector to reliably and efficiently identify fraudulent credit card transactions. Conventional detection approaches fail to adapt to the advancing strategies of fraudsters, resulting in heightened false positives and inefficiency within fraud detection systems. This study overcomes these restrictions by creating an innovative stacking hybrid machine learning (ML) approach that combines decision trees (DT), random forests (RF), support vector machines (SVM), XGBoost, CatBoost, and logistic regression (LR) within a stacking ensemble framework. This method uses stacking to integrate diverse ML models, enhancing predictive performance, with a meta-model consolidating base model predictions, resulting in superior detection accuracy compared to any single model. The methodology utilizes sophisticated data preprocessing techniques, such as correlation-based feature selection and principal component analysis (PCA), to enhance computing efficiency while preserving essential information. Experimental assessments of a credit card transaction dataset reveal that the stacking ensemble model exhibits higher performance, achieving an F1-score of 88.14%, thereby efficiently balancing precision and recall. This outcome highlights the significance of ensemble methods such as stacking in attaining strong and dependable cyber fraud detection, emphasizing its capacity to markedly enhance the security of financial transactions.

Corresponding author
Eyad Abdel Latif Marazqah Btoush,
EyadAbdelLatif.A.Q.
MarazqahBtoush@usq.edu.au

## INTRODUCTION

Credit cards have become the preferred payment method by virtue of the exponential development of online transactions, which has resulted in an estimated global transaction volume exceeding $20 trillion annually. However, this expansion has also propelled an increase in cyber fraud, which results in financial institutions incurring billions of dollars in annual losses. Recent projections indicate that the global losses from credit card fraud exceeded $32 billion in 2023 and could increase to $43 billion by 2026 if effective measures are not implemented (*Merchant Cost Consulting, 2023*). This ongoing obstacle has necessitated the urgent implementation of sophisticated fraud detection systems in the

financial sector. These systems must be capable of combating the increasing sophistication of fraudulent strategies, which compromise consumer trust and result in substantial financial losses on a global scale.

Machine learning (ML) techniques to resolve these challenges have been demonstrated through their capacity to process large datasets, capture complex patterns, and provide superior predictive accuracy (*Btoush et al., 2023*). These methods have been demonstrated to be effective in distinguishing between legitimate and fraudulent activities by learning from historical transaction data. Nevertheless, the detection of fraud is still a complex process due to the computational requirements of analyzing vast transaction data and the highly imbalanced datasets. Models that are biased toward predicting legitimate transactions are frequently the result of the scarcity of fraudulent instances, which increases the probability of missing fraud cases.

Among ensemble learning strategies, stacking has emerged as a particularly promising solution due to its theoretical advantage of meta-model synthesis. By combining the outputs of multiple heterogeneous base learners, stacking enables the meta-learner to capture complementary predictive patterns, reducing both model bias and variance (*Dal Pozzolo et al., 2014*; *Abdelghafour et al., 2024*). Unlike simpler ensemble approaches such as bagging or boosting, stacking facilitates more sophisticated integration of diverse model predictions, making it well-suited for complex, imbalanced fraud datasets.

A plausible solution has emerged in the form of ensemble learning approaches, particularly stacking, which combine multiple models to capitalize on each model's unique strengths while minimizing individual weaknesses. In environments where the data is diverse and complex, this strategy is particularly advantageous for improving predictive performance.

This study is motivated by the requirement for a sophisticated, scalable fraud detection system that can improve the accuracy and robustness of credit card fraud detection. Our present research work suggests a composite stacking ensemble model that incorporates a variety of ML algorithms, such as decision trees (DT), random forests (RF), support vector machines (SVM), XGBoost, CatBoost, and logistic regression (LR). The selection of these models is strategic in nature, as RF and DT are particularly adept at managing imbalanced data and offer interpretability. LR functions as a benchmark model that provides a simple classification approach, enabling a comparative analysis of more intricate methodologies. Additionally, SVM is incorporated due to their ability to effectively manage high-dimensional, non-linear data. The selection of XGBoost and CatBoost is based on their exceptional capabilities in managing large datasets with categorical features. This makes them particularly effective in identifying subtle fraud patterns that conventional models may overlook.

Our stacking ensemble enhances detection performance by aggregating predictions from these diverse models through a meta-model, surpassing the capabilities of any particular model. The meta-model effectively integrates the assets of the individual base learners. We employ data preprocessing techniques, including principal component analysis (PCA) and correlation-based feature selection, to further enhance efficiency and accuracy. This study emphasizes the potential of ensemble learning to improve the security

of financial transactions by introducing a fraud detection model that is adaptable and robust, and that is capable of overcoming the intricate and dynamic challenges associated with cyber fraud. In accordance with the changing financial cybersecurity landscape, this present research work is a fundamental step in the development of more effective fraud detection systems that can adapt to the evolving strategies of cybercriminals, thereby enhancing the overall resilience of the financial sector.

The remainder of this present research work is organized as follows. 'Related Work' presents the related work, and 'Methodology' presents the methodology used in this study. 'Proposed Stacking Hybrid ML' introduces the proposed stacking model. 'Results and Discussion' presents the results and discussion, and 'Conclusion' presents the conclusions.

## RELATED WORK

Machine learning (ML) has become a potent tool for identifying anomalous behaviors and predicting risks, particularly in the domain of credit card fraud detection (*Btoush et al., 2024*). Its capacity to analyze large-scale, high-dimensional datasets makes ML highly suitable for recognizing fraudulent patterns in transactional data (*Ozkan-Ozay et al., 2024*).

Supervised learning approaches, which rely on labeled datasets, are central to fraud detection efforts. Among them, classification algorithms are critical for determining whether a transaction is legitimate or fraudulent (*Gupta et al., 2023*). *Reddy & Sriramya (2023)* compared SVM and DT, reporting higher accuracy for SVM (98.59%) than DT (94.86%). *Nama, Obaid & Alrammahi (2023)* enhanced SVM performance with multilayer perceptron (MLP), while SVM retained a marginal lead. *Mukherjee et al. (2021)* demonstrated that DTs can achieve up to 99% accuracy, and *Amusan et al. (2021)* showed that RF outperformed other classifiers on distorted data. *Hema & Muttipati (2020)* further improved results by combining RF with CatBoost, achieving 99.5% accuracy, while *Ileberi, Sun & Wang (2022)* achieved 99.98% accuracy using a genetic algorithm–RF hybrid model.

Beyond individual classifiers, ensemble techniques have emerged as powerful solutions due to their ability to integrate multiple model predictions. Among these, stacking is a prominent approach that combines the outputs of several base learners into a meta-classifier. *Faraj, Mahmud & Rashid (2021)* found that XGBoost outperformed other ensemble models in credit card cyber fraud prediction. Similarly, *Muaz, Jayabalan & Thiruchelvam (2020)* reported improved detection rates using layered ensemble architectures.

Several studies have explored stacking-based models in greater depth. *Awoyemi, Adetunmbi & Oluwadare (2017)* combined LR, RF, and gradient boosting, reporting enhanced precision despite higher computational demands. *Dal Pozzolo et al. (2014)* emphasized that while stacking improves rare-event prediction, it requires careful resampling to mitigate bias. However, most existing works do not rigorously justify their meta-learner choices and often rely on limited combinations of base classifiers, reducing ensemble diversity and limiting generalizability.

Recent studies have introduced advanced stacking models with robust designs. For instance, a study published in Results in Engineering by *Gupta et al. (2025)* integrated Synthetic Minority Oversampling Technique and Edited Nearest Neighbors (SMOTE-ENN), autoencoders, and a particle swarm optimization (PSO)-optimized stacking ensemble, achieving 99.97% accuracy, 99.59% precision, and 99.9% recall on a real-world dataset. However, their reliance on synthetic sampling may introduce artificial patterns, potentially compromising real-world validity. Similarly, *Chagahi et al. (2024)* proposed an attention-based ensemble system that achieved 99.95% accuracy and an area under the curve (AUC) of 1. While the model demonstrates high performance, its complexity may hinder real-time deployment.

Additionally, many recent models continue to use LR or deep learning models as meta-learners by default, without evaluating alternative structures like tree-based learners. This is problematic, as LR may not effectively capture non-linear relationships, especially in complex, high-dimensional fraud data. Furthermore, base learners in many stacking studies often stem from similar algorithm families, reducing the ensemble's ability to model diverse fraud patterns. Limited metric evaluations and overreliance on synthetic techniques like SMOTE also restrict the practical deployment of these models.

This present research work addresses these limitations through a diversified stacking ensemble framework that incorporates DT, RF, SVM, XGBoost, CatBoost and LR as base learners, capturing both tree-based and margin-based learning paradigms. It employs RF as the meta-learner, diverging from the common reliance on LR, and offering superior performance in modeling complex, non-linear decision boundaries.

Crucially, this approach avoids over-dependence on synthetic oversampling methods like SMOTE, instead emphasizing robust model architecture and multi-metric evaluation (precision, recall, F1-score, and AUC-ROC) to address data imbalance. The proposed ensemble is computationally efficient, interpretable, and scalable, making it suitable for real-world deployment where fraudulent instances are rare yet consequential.

Overall, this work contributes a more generalizable and practically viable stacking strategy that bridges the gap between academic experimentation and deployment-ready fraud detection systems.

## METHODOLOGY

This section explores the credit card dataset employed in the study and provides a comprehensive explanation of the various algorithms and strategies utilized in formulating the suggested credit card cyber fraud detection methodology.

### Dataset

To assess the effectiveness of the suggested ML models, a widely recognized dataset was chosen for both training and testing. This dataset is available at https://www.kaggle.com/mlg-ulb/creditcardfraud. The dataset consisted of client transactions at a European bank in 2013. The real-world dataset consists of 284,807 credit card transactions.

### Programming language

Python, an interpreted and high-level programming language, was used to develop the system. NumPy 1.26.0 and Pandas 2.1.1 handled data manipulation, while Scikit-learn 1.3.2 supported preprocessing, PCA, and machine learning models. Imbalanced-learn 0.11.0 addressed class imbalance, and CatBoost 1.2, XGBoost 1.7.6, and LightGBM 4.1.0 implemented gradient boosting methods. Data visualization used Seaborn 0.13.0. All work was conducted using Python 3.11.4 in Jupyter Notebook on a 3.3 GHz Intel Core i7 machine with 16 GB RAM.

### Classification techniques

Multiple ML models were trained using RF, DT, LR, SVM, XGBoost, and CatBoost algorithms on the dataset. After training the model, its performance was evaluated, and a visual representation was created to show the differences between the algorithms.

### Evaluation and reflection

This present research work evaluates machine learning algorithms for cyber fraud detection using five-fold cross-validation, a robust approach for mitigating class imbalance and ensuring generalizable results. Model performance was primarily assessed using the F1-score, which balances precision and recall. Additional evaluation metrics, including accuracy, confusion matrix, recall, precision, and Area Under the Receiver Operating Characteristic (AUC-ROC), were employed to offer a comprehensive understanding of each model's selectivity, specificity, and overall predictive power. To rigorously validate the observed performance differences between the proposed stacking ensemble and individual base classifiers, paired t-tests were conducted across the cross-validation folds. In terms of computational efficiency, training time was measured for each model to assess the practicality of deploying complex ensembles.

## PROPOSED STACKING HYBRID ML

This section describes the successful development of a hybrid ML approach that integrates DT, RF, SVM, XGBoost, CatBoost, and LR with an ensemble learning technique. This approach addresses the current challenges of detecting cyber fraud in credit card transactions by combining numerous base models. Stacking is an ensemble learning technique that combines multiple base models to improve predictive performance. In this implementation, base models include diverse ML algorithms. The meta-model (or final estimator) makes the ultimate prediction by aggregating predictions from the base models. The meta-model used in this approach is RF. The hyperparameters for this meta-model are chosen to optimize performance, including:

*n_estimators = 100 (number of trees in the forest)*

*max_depth = 10 (limits tree depth to prevent overfitting)*

*min_samples_split = 2 (minimum number of samples to split a node)*

*min_samples_leaf = 1 (minimum samples required at a leaf node)*

*random_state = 42 (to ensure reproducibility).*

The stacking process begins by training the base models (DT, RF, SVM, XGBoost, CatBoost, and LR) independently on the dataset. Each base model outputs predictions for the test set. These predictions are then fed as features into the meta-model, which makes the final prediction based on these aggregated outputs.

*# Step 1: Train base models*

*base_model1 = DT().fit(X_train, y_train)*

*base_model2 = RF().fit(X_train, y_train)*

*base_model3 = SVM().fit(X_train, y_train)*

*base_model4 = XGBoost().fit(X_train, y_train)*

*base_model5 = CatBoost().fit(X_train, y_train)*

*base_model6 = LR().fit(X_train, y_train)*

*# Step 2: Make predictions using base models*

*predictions_base_models = [*
    *base_model1.predict(X_test),*
    *base_model2.predict(X_test),*
    *base_model3.predict(X_test),*
    *base_model4.predict(X_test),*
    *base_model5.predict(X_test),*
    *base_model6.predict(X_test)]*

*# Step 3: Combine predictions into a new dataset for the meta-model*

*stacked_features = np.column_stack(predictions_base_models) # Combine base model predictions*

*# Step 4: Train meta-model (random forest)*

*meta_model = RF(n_estimators = 100, max_depth = 10, min_samples_split = 2, min_samples_leaf = 1, random_state = 42)*

*meta_model.fit(stacked_features, y_test) # Train meta-model on stacked predictions*

*# Step 5: Make final prediction using meta-model*

*final_prediction = meta_model.predict(stacked_features) # Meta-model predicts final output*

The present research work applies the publicly available credit card transaction dataset to train the algorithm and subsequently detect cyber fraud. To attain this, a

correlation-based feature selection technique was implemented to identify and extract the most relevant features for predicting the target variable. To improve the computational efficiency of the algorithm, the full dataset was reduced to only the most significant input features by applying principal component analysis (PCA), and the outputs of the selected features were then transferred to the new model that combines ML techniques using stacking ensemble techniques.

The stacking arrangement is theoretically motivated by the principle of combining diverse learners to maximize generalization. Tree-based models (DT, RF, XGBoost, CatBoost) capture non-linear relationships and feature interactions, SVM is effective for margin-based separation, and logistic regression adds probabilistic reasoning. Their combined predictions are fed into a random forest meta-learner, which benefits from ensemble aggregation and is capable of modeling complex patterns without overfitting. RF also handles noisy or redundant base-level outputs well, making it particularly suited for imbalanced and high-dimensional fraud detection problems.

The novelty of this present research work lies in the development of a hybrid ML model capable of simultaneously performing feature extraction and classification to distinguish between cyber fraud and non-fraud transactions. This present research work contributes by developing efficient algorithms for dimensionality reduction that prioritize key features using correlation-based analysis, random forest, XGBoost, and permutation feature importance methods. It also introduces a stacking hybrid ML approach to automatically classify transactions as fraud or not. Algorithm 1 presents the stacking hybrid ML model. This model integrates both feature extraction and classification steps into a unified stacking procedure to achieve accurate detection of credit card fraud.

This study proposes a novel stacking ensemble framework that combines six diverse base learners and uses random forest as a meta-classifier. This configuration allows modeling both linear and complex nonlinear patterns while maintaining robustness. Unlike many existing works that rely on logistic regression at the meta-level, our use of a tree-based learner adds modeling depth. Furthermore, we apply targeted preprocessing and prioritization of fraud-relevant evaluation metrics such as F1-score and AUC-ROC, avoiding over-reliance on synthetic resampling. These design choices distinguish our method from prior stacking-based fraud detection systems and contribute to its superior performance in detecting fraudulent transactions. Figure 1 illustrates a block diagram of the proposed modeling framework developed in this article.

## Data processing

A dataset of 284,807 real-world credit card transactions was used to develop the cyber fraud detection model. It contains 31 columns: 30 features and one target class indicating whether a transaction is fraudulent or genuine. The dataset shows a significant class imbalance, with only 492 fraudulent transactions (0.173%) and 284,315 non-fraudulent transactions (99.827%). Figure 2 shows the percentage of fraudulent *vs*. non-fraudulent transactions. Due to this imbalance, raw data may not yield accurate results. The dataset

**Algorithm 1** Stacking hybrid ML model.

```
1: Procedure StackingClassifierTraining (X, y, test_size, random_state)
2:    Pre-process (X, y)
4:    Split dataset into (X_train, X_test, y_train, y_test)
5:    Normalize (X_train, X_test)
6:    Model ← Create stacking classifier with RandomForestClassifier
             as meta-classifier()
7:          ('Random Forest', rf_classifier)
8:          ('SVM', svm_classifier)
9:          ('Logistic Regression', lr_classifier)
10:         ('Decision Tree', dt_classifier)
11:         ('XGBoost', xgb_classifier)
12:         ('CatBoost', catboost_classifier)
13:          Cross-validation: StratifiedKFold with cv_folds
14: for k ← 0 to n-1 do
15:    Train StackingClassifier (X_train, y_train)
16:    Predict y_pred on (X_test)
17:    Evaluate Model:
18:       Accuracy ← accuracy_score(y_test, y_pred)
19:       Precision ← precision_score(y_test, y_pred)
20:       Recall ← recall_score(y_test, y_pred)
21:       F1-score ← f1_score(y_test, y_pred)
22:       ROC AUC Score ← roc_auc_score(y_test, y_pred)
23:    Print Evaluation Metrics
24: end for
25: End Procedure
```

provides anonymized and transformed feature values; specifically, the original attributes have been processed using PCA by the dataset creators. The pre-processing tasks have been accomplished by utilizing the Python data manipulation package pandas and the machine learning module Scikit-learn. The sequential process is visually depicted in Fig. 3.

### Data cleaning

Python imported the credit card dataset using the proper import command. A thorough data cleansing followed. Two main data cleansing activities are typically used. The first step is removing null and absent values from the dataset. Management of outliers—data points that differ significantly from the majority—is the second responsibility. The dataset has 284,807 transactions. Null values were absent from the dataset imported the credit card dataset using the proper import command. A thorough data cleansing followed. The first step is removing null and absent values from the dataset. Management of outliers is the second responsibility.

*Peer*J Computer Science

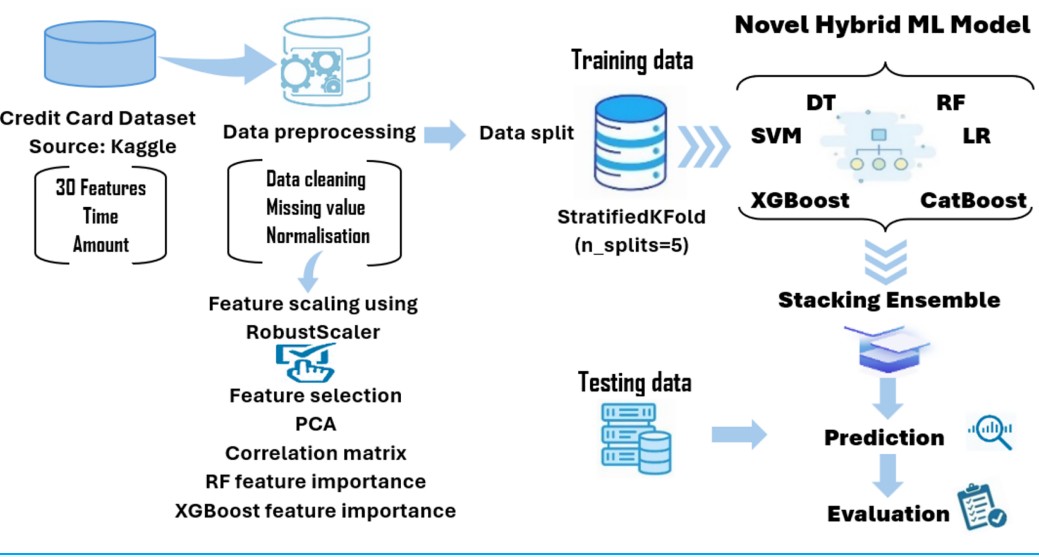

**Figure 1  The novel hybrid stacking ML model.** 

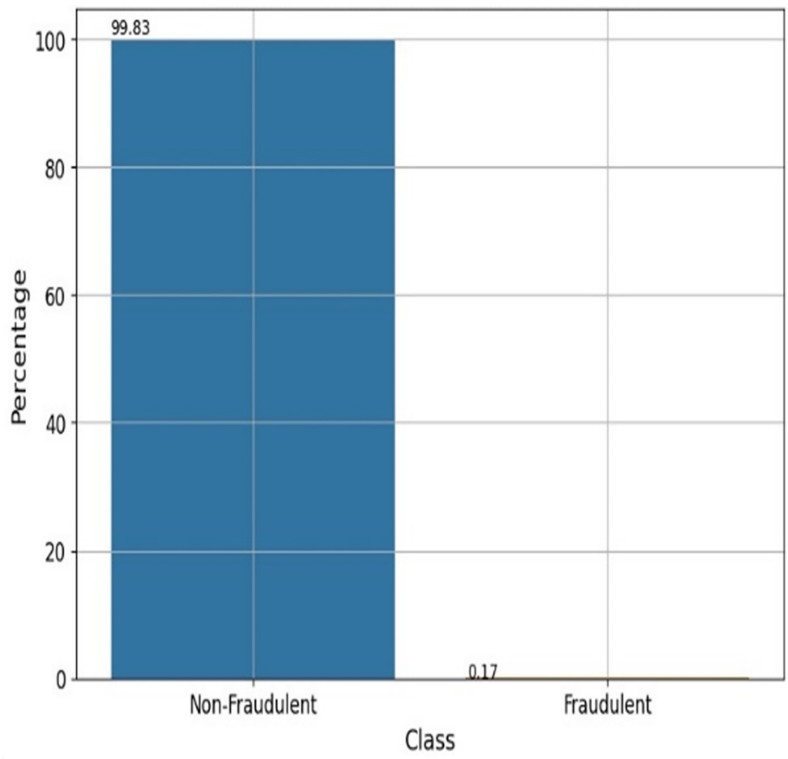

**Figure 2  Fraudulent *vs* non-fraudulent transactions.**

## Feature scaling

Data pre-processing includes the step of normalizing a dataset's independent variables to ensure consistent scales across features. This process centers the data around 0 or scales it between 0 and 1, depending on the scaling method used. Feature scaling improves model

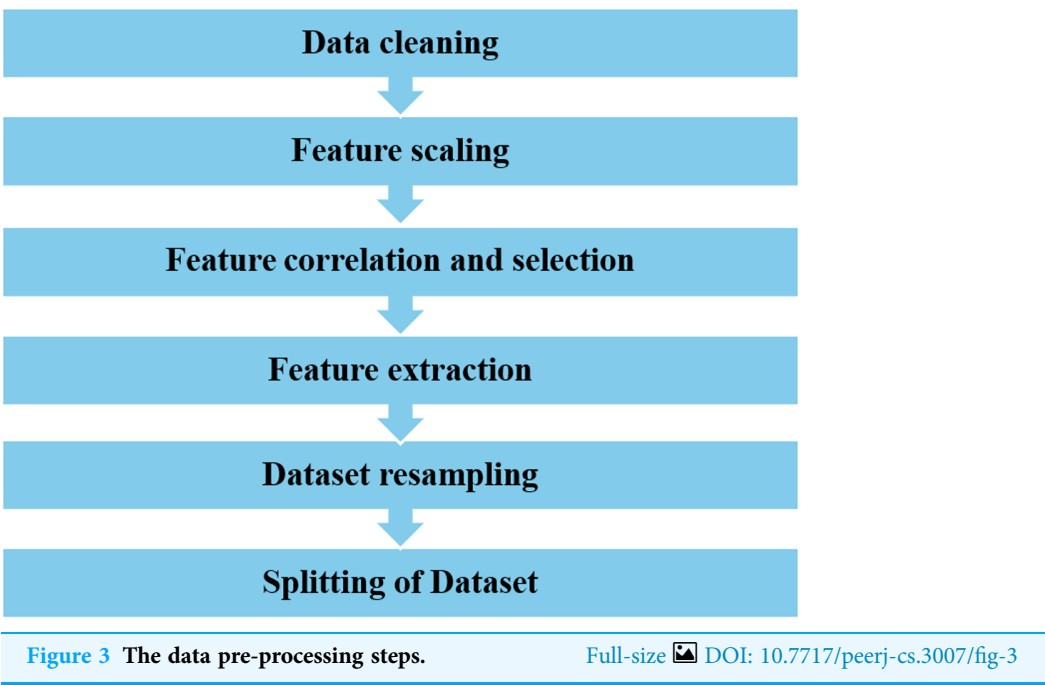

**Figure 3** The data pre-processing steps.     

performance by ensuring that all features contribute equally to the model's learning process. In this study, we utilized the existing Standard Scaling technique. The StandardScaler was applied to the training data (X_train) and the test data (X_test) to scale their features accordingly.

### Feature correlation and selection

Feature selection is essential for enhancing model performance, reducing computational complexity, and improving interpretability. In this study, we combined statistical correlation analysis and machine learning-based feature importance techniques to systematically identify the most relevant features for cyber fraud detection.

Initially, Pearson correlation coefficients were calculated to assess the linear relationships between features and the target variable 'Class'. Although the principal components (V1–V28) were generally uncorrelated with each other, several exhibited strong associations with the target. To further refine feature selection, feature importance scores were computed using random forest, XGBoost, and permutation importance methods. The results from these analyses were consolidated and visualized in Fig. 4.

Seventeen features were ultimately selected (V17, V14, V12, V10, V16, V3, V7, V11, V4, V18, V1, V9, V5, V2, V6, V21, and V19) based on their consistently high importance across different methods. The choice of 17 features was further validated through internal comparisons, where models using 10, 17, and all 30 features were evaluated. Using 17 features provided the best balance between predictive accuracy, generalization ability, and computational efficiency. This rigorous selection process ensured that the final model was both effective and interpretable.

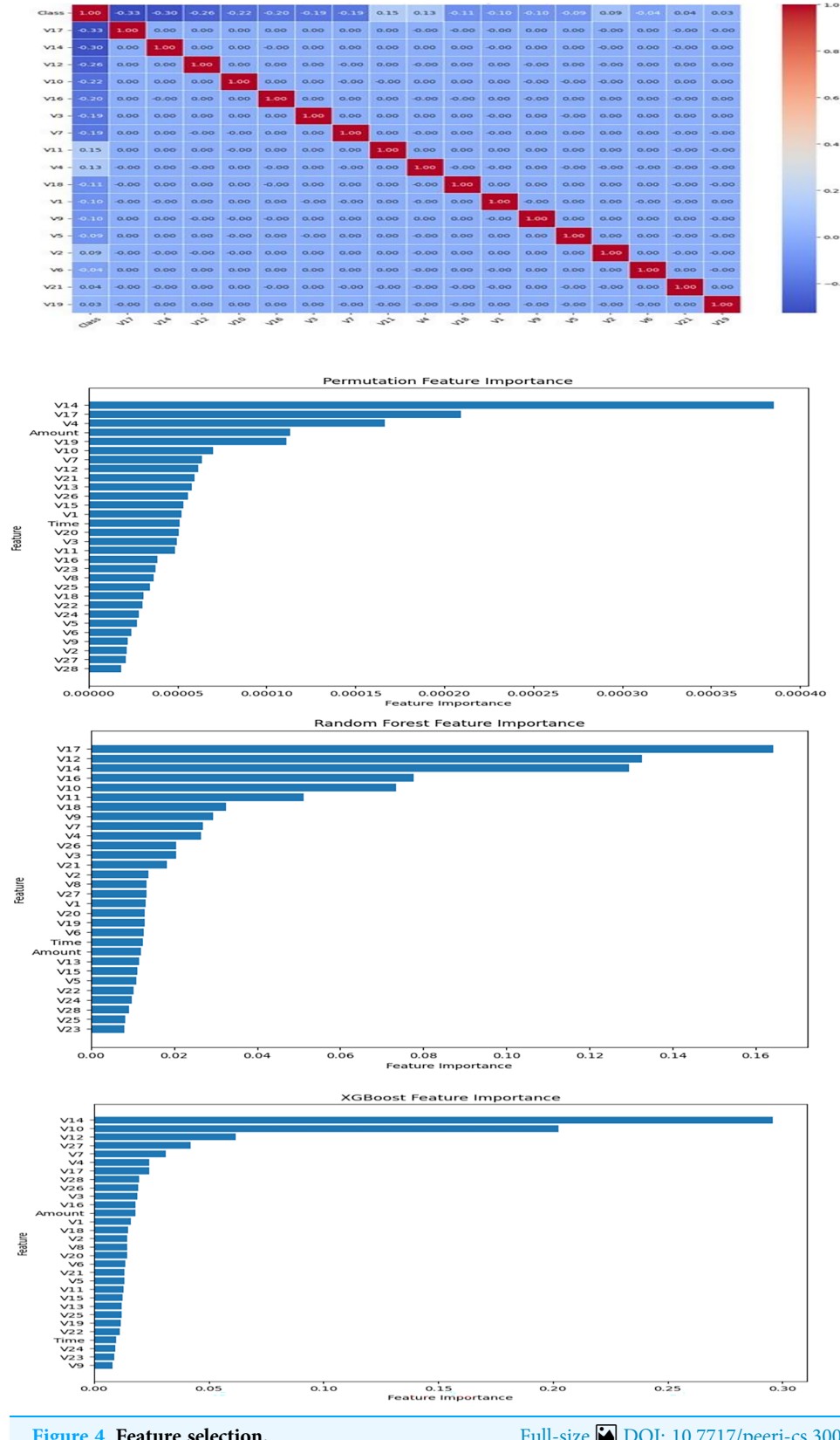

**Figure 4** **Feature selection.**

### Feature extraction

Feature extraction is a critical pre-processing step in ML that transforms data into a set of informative features, thereby improving model performance, efficiency, and interpretability. Techniques like PCA are used for dimensionality reduction, helping to reduce noise, prevent overfitting, and lower computational costs. Although the dataset features were already transformed through PCA by the original providers, we applied an additional PCA using Scikit-learn's default parameters to further reduce dimensionality for improved model performance and visualization. PCA demonstrated acceptable discrimination between classes and was thus selected as the feature extraction technique for subsequent training and testing phases.

### Data splitting

This method serves two crucial purposes: mitigating the risk of overfitting and verifying the performance of the model in real-world situations. The complete dataset is partitioned into a training set comprising 80% of the data and a test set including the remaining 20%.

### Hyperparameter tuning

Following data pre-processing, hyperparameter tuning was conducted using GridSearchCV. The grid search was applied with 5-fold cross-validation on the training data to optimize model performance while avoiding overfitting. For each machine learning algorithm (DT, RF, SVM, XGBoost, CatBoost, and LR), key hyperparameters were tuned across predefined ranges, including learning rate, number of estimators, maximum depth, and regularization parameters, depending on the model. Specifically, the ranges considered for hyperparameters like n_estimators, max_depth, learning_rate, C, and subsample were systematically tested. The best-performing hyperparameter combinations were then used to retrain the models.

## Machine learning techniques

### Decision tree

DT is a supervised ML algorithm used for classification tasks. In this specific implementation, the DT classifier is configured with the following hyperparameters; 'max_depth': 70, 'min_samples_split': 8, 'min_samples_leaf': 10, and 'criterion': "entropy". The 'max_depth' parameter controls the maximum depth of the decision tree, limiting its complexity and preventing overfitting. 'Min_samples_split' specifies the minimum number of samples required to split an internal node, while 'min_samples_leaf' sets the minimum number of samples required to be at a leaf node. These parameters help regulate the size of the tree and improve its generalization capability. Additionally, the 'criterion' parameter determines the function used to measure the quality of a split. In this case, "entropy" is chosen, which computes the information gain based on the entropy.

### Random forest

RF is a powerful ensemble learning technique that builds multiple decision trees and outputs their classification mode or regression mean. Key hyperparameters include n_estimators (set to 100 to reduce overfitting risk), max_depth (limited to 10 to control

complexity and prevent overfitting), min_samples_split (set to 2 for balanced splits), and min_samples_leaf (set to 1 to prevent low-sample nodes). A random_state of 42 ensures reproducibility. These hyperparameters balance generalization and model complexity, effectively capturing data patterns without overfitting. Adjustments should consider dataset characteristics and the bias-variance trade-off.

### Support vector machine

SVM is a popular supervised learning type. A dataset separates features (X) and target attribute (y). Data for model evaluation is 80/20 training and testing. A hyperparameter-based SVM start kernels: SVM kernels provide a higher-dimensional space with more input-data separable classes. RBF kernel 'rbf'. Helpers: Support vectors move the decision border near the hyperplane. Main causes limit choice. Data point margin is decision boundary-to-nearest class. SVM margins reduce overfitting and demonstrate classifier prediction confidence. Category margin and error are limited by regularization parameter 'C'. Making 'C' 1.0 balances goals. Individual training examples alter gamma. Automatically calculate 1/(n_features * X.var ()) with 'gamma' set to 'scale' for training data. Values match data kernel coefficient scale. SVMs predict by labelling new data points by decision boundaries. Hyperplane data point signed distance is calculated by a binary classification decision function. Negative and positive distances distinguish classes. When classifying binary data, SVM maximizes feature space margin.

### XGBoost

Gradient boosting, specifically XGBoost, builds decision trees that iteratively improve each other to classify binary events. The objective function, set to 'binary: logistic,' uses logloss to evaluate model performance by penalizing inaccurate predictions. The learning rate ('eta') controls the step size for each iteration, with a lower value preventing overfitting but requiring more iterations. The 'max_depth' parameter limits tree complexity by setting the depth to 6. Subsampling training instances and features (with values of 0.8 for 'subsample' and 'colsample_bytree') improves model performance and reduces overfitting. A random seed of 42 ensures reproducibility of results.

### CatBoost

CatBoost is a powerful gradient boosting algorithm that efficiently handles categorical variables internally, eliminating the need for one-hot or label encoding and reducing data pre-processing errors. Its default 'Logloss' objective function achieves high accuracy. Key hyperparameters—iterations, learning_rate, depth, and l2_leaf_reg—allow for fine-tuning. Iterations control the number of boosting rounds; learning_rate adjusts weight changes for smooth convergence; depth balances complexity; and l2_leaf_reg applies regularization to prevent overfitting. These settings allow customization of CatBoost to match data characteristics and optimization goals.

### Logistic regression

LR is an effective classification method, particularly suitable for binary classification tasks. The 'liblinear' solver, chosen for small to medium datasets, efficiently manages model

complexity and multicollinearity through L1 and L2 regularization. L1 regularization (Lasso) promotes model sparsity by penalizing large coefficients, simplifying feature selection and enhancing model interpretability. L2 regularization (Ridge) prevents overfitting by limiting the growth of coefficients. To ensure reproducibility, the 'random_state' is set to 42, enabling consistent model behavior for debugging, validation, and comparison.

## RESULTS AND DISCUSSION

Initially, we assess the performance of ML algorithms individually and in the absence of employing ensemble techniques. The outcomes derived from this evaluation are comprehensively depicted in Table 1, illustrating a comparative analysis of the algorithms. Figure 5 shows the performance of ML algorithms without ensemble techniques. Figures 6, 7, 8, 9, 10, and 11 show the confusion matrix for ML techniques.

### Comparison between ML techniques before ensemble

The decision tree (DT) model demonstrates high accuracy (99.93%) and precision (89.89%), but its recall (81.63%) for fraudulent transactions indicates that it may miss some instances of fraud, reflected in its F1-score of 85.56%. The recall suggests that the model could improve in detecting all instances of fraud. The AUC-ROC score of 0.9079 indicates a good ability to discriminate between fraudulent and non-fraudulent transactions, though there's room for improvement in capturing more positive instances.

The RF model achieves exceptional accuracy (99.96%) and precision (97.40%), but its recall (76.53%) for fraudulent transactions is lower, meaning it misses a substantial number of fraudulent transactions. The F1-score of 85.71% shows a balanced performance. Its AUC-ROC score of 0.9725 is excellent, highlighting its strong discrimination ability and suggesting that RF is good at distinguishing between the two classes, but improving recall could enhance its fraud detection capability further.

The SVM model achieves 99.94% accuracy and 97.02% precision, but its recall of 66.33% for fraudulent transactions results in a lower F1-score of 78.79%. This lower recall suggests that SVM struggles to capture fraudulent transactions, and its AUC-ROC score of 0.9513 supports this, showing that while SVM is effective, it is not as strong at differentiating between classes, particularly for fraud detection.

XGBoost excels with 99.95% accuracy and 95.00% precision, with a recall of 77.55% for fraudulent transactions, which is higher than SVM and RF in detecting fraudulent transactions. Its F1-score of 85.39% shows a strong balance between recall and precision. The AUC-ROC score of 0.9783 is also very high, indicating that XGBoost has a strong ability to discriminate between fraudulent and non-fraudulent transactions. While not the top performer in recall, its strong AUC-ROC score and solid F1-score make it a reliable model for fraud detection.

CatBoost performs excellently with 99.96% accuracy, 97.46% precision, and a strong recall of 77.55% for fraudulent transactions, leading to the highest F1-score of 86.35%. Its AUC-ROC score of 0.9837 is the highest among all models, indicating superior

**Table 1 Algorithm performance.**

| ML | Accuracy (%) | Precision (%) | Recall (%) | F1-score (%) | AUC (%) | *p*-value (*vs* Stacking) | 95% CI of F1 difference |
|---|---|---|---|---|---|---|---|
| DT | 99.93 | 89.89 | 81.63 | 85.56 | 90.80 | 0.2027 | [−0.0161 to 0.0550] |
| RF | 99.96 | 97.40 | 76.53 | 85.71 | 97.25 | 0.0199 | [−0.0450 to −0.0067] |
| SVM | 99.94 | 97.02 | 66.33 | 78.79 | 95.13 | 0.0225 | [0.0083–0.0635] |
| XGBoost | 99.95 | 95.00 | 77.55 | 85.39 | 97.83 | 0.0628 | [−0.0261 to 0.0011] |
| CatBoost | 99.96 | 97.44 | 77.55 | 86.36 | 98.37 | 0.0315 | [−0.0423 to −0.0033] |
| LR | 99.92 | 88.06 | 60.20 | 71.52 | 97.01 | 0.0047 | [0.0593–0.1727] |

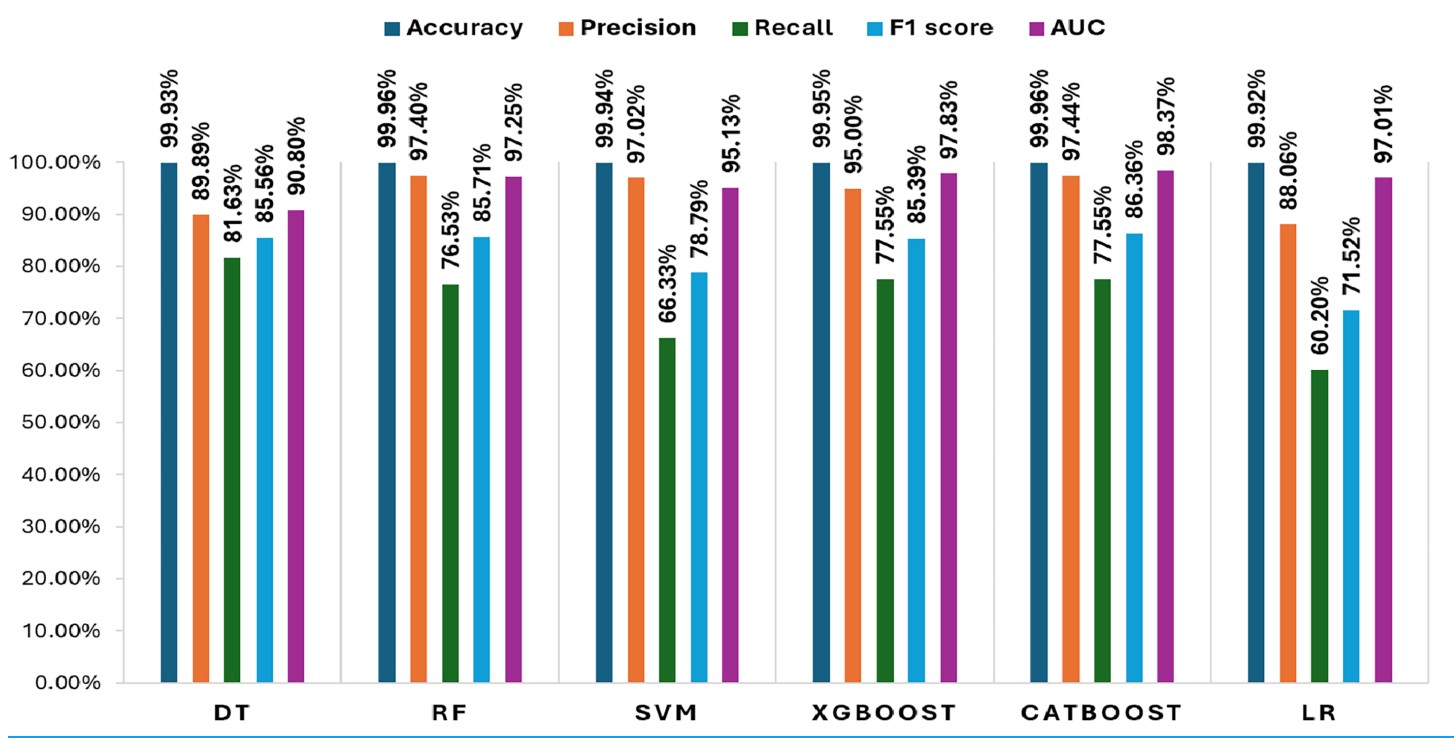

**Figure 5 Algorithms performance before applying ensemble techniques.**

ability to discriminate between the classes. The high precision, recall for fraudulent transactions, and AUC-ROC score make CatBoost the top performer in fraud detection, providing an excellent balance in identifying fraudulent transactions without sacrificing precision.

LR achieves high accuracy (99.92%) but has lower precision (88.06%) and recall (60.20%) for fraudulent transactions, leading to a moderate F1-score of 71.52%. Its AUC-ROC score of 0.9701 further suggests that while LR performs well in terms of accuracy, it struggles to distinguish fraudulent transactions, especially when compared to other models. The low recall and AUC-ROC indicate that LR is less effective at identifying the minority class of fraud.

Computer Science

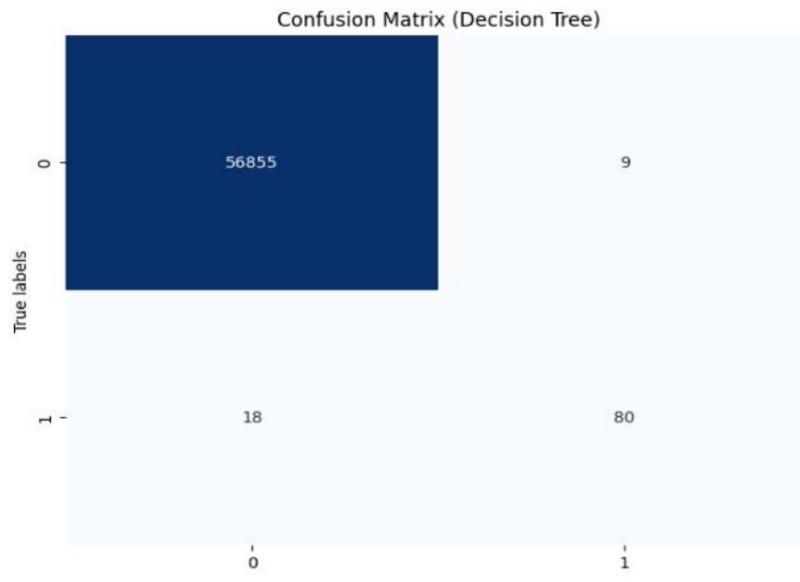

Figure 6 Confusion matrix-DT.               

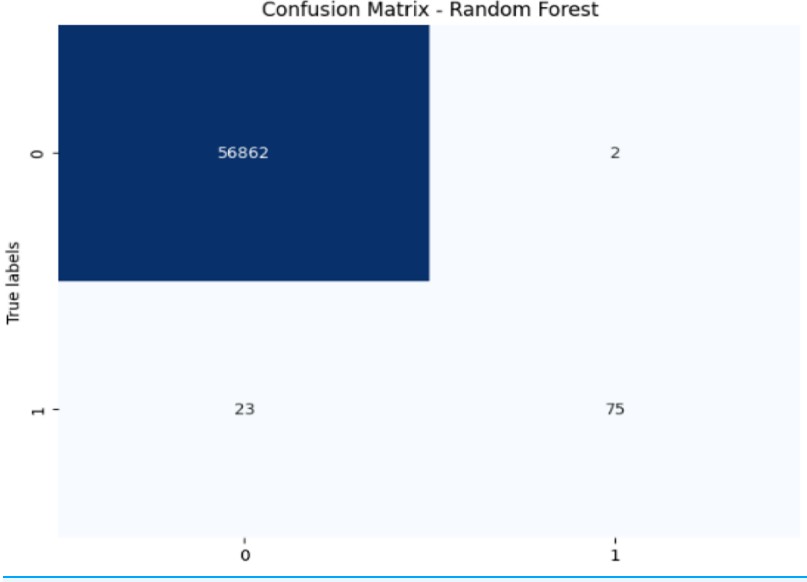

Figure 7 Confusion matrix-RF.               

All models exhibit high accuracy, but their performance differs when considering precision, recall for fraudulent transactions, F1-score, and AUC-ROC. Given the importance of detecting fraudulent transactions, recall is particularly critical, and F1-score is prioritized as it provides a balanced assessment of precision and recall. The AUC-ROC scores provide an additional measure of how well each model can discriminate between classes.

In terms of performance, CatBoost leads with the highest F1-score of 86.35%, followed by random forest (85.71%) and XGBoost (85.39%), all of which show excellent balance

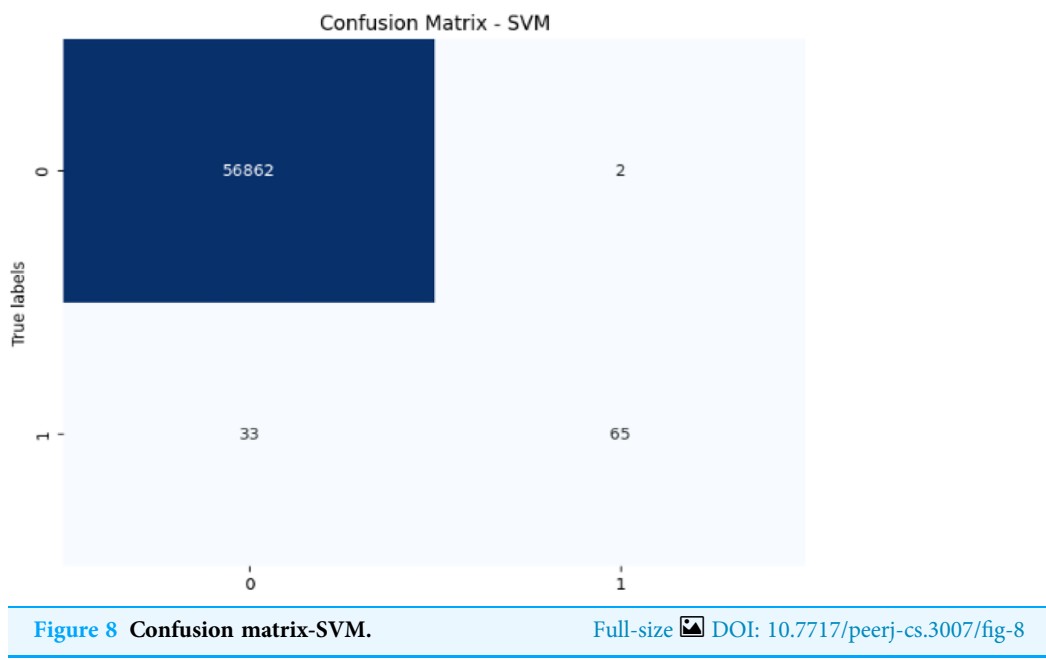

**Figure 8 Confusion matrix-SVM.**    

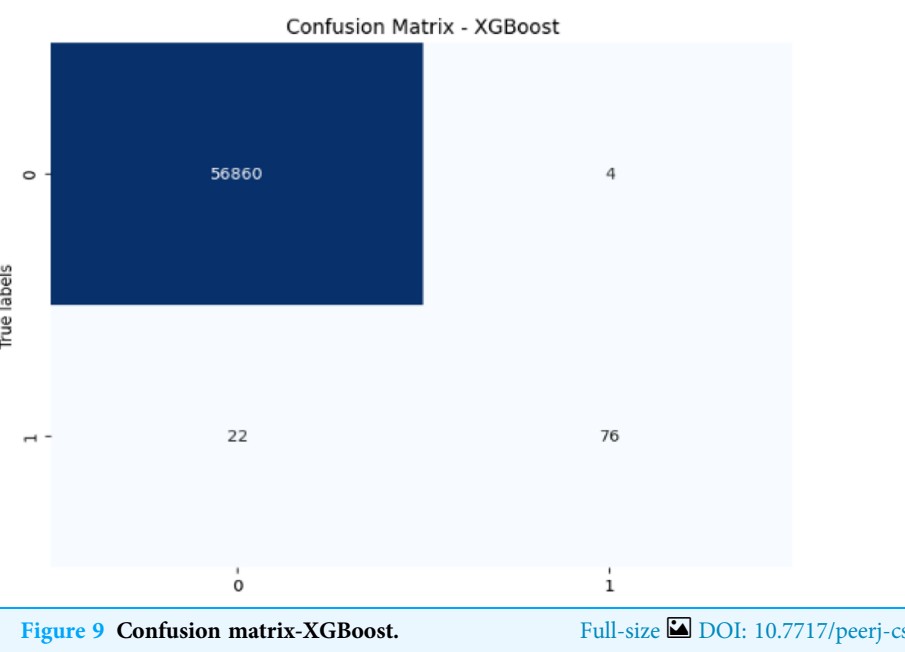

**Figure 9 Confusion matrix-XGBoost.** 

between precision and recall. Their AUC-ROC scores (CatBoost: 0.9837, RF: 0.9725, XGBoost: 0.9783) reinforce their strong ability to discriminate between fraudulent and non-fraudulent transactions. DT (85.56%) also performs well, with an AUC-ROC score of 0.9079, though it lags behind in recall. Meanwhile, SVM (78.79%) and LR (71.52%) have lower F1-scores and AUC-ROC scores, reflecting their challenges in identifying fraudulent transactions.

Overall, CatBoost and random forest stand out as the top performers, with CatBoost also leading in terms of AUC-ROC score. These models offer strong fraud detection

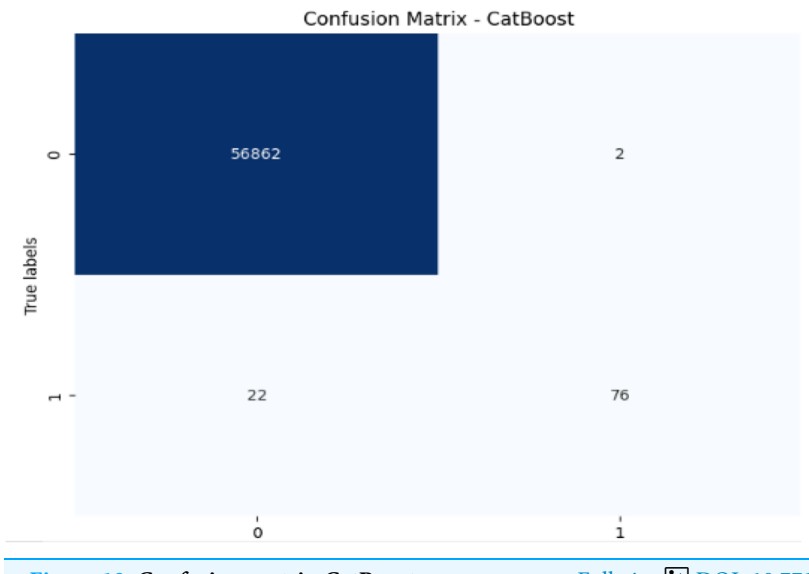

**Figure 10  Confusion matrix-CatBoost.**

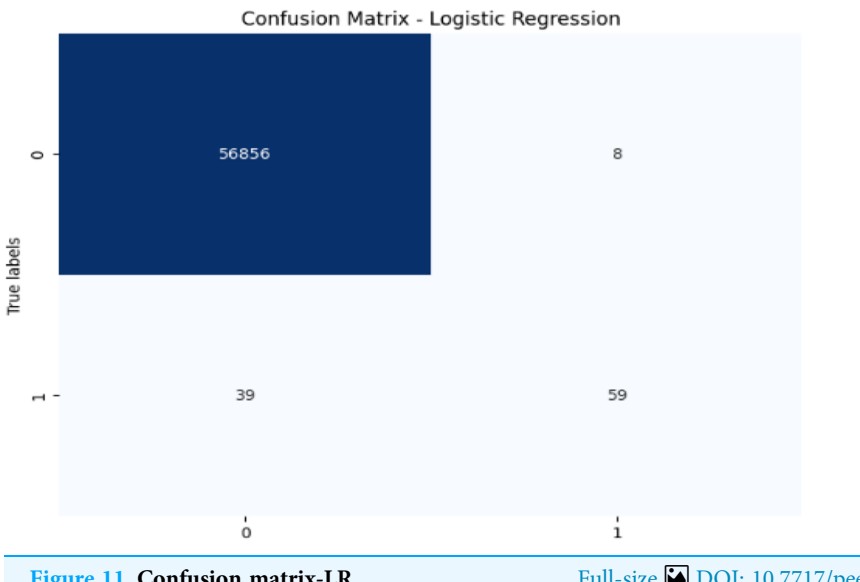

**Figure 11  Confusion matrix-LR.**

capabilities, balancing recall and precision effectively. Figure 12 illustrates the F1-scores for the various machine learning techniques, while the AUC-ROC scores further confirm the models' ability to differentiate between fraudulent and non-fraudulent transactions.

### Ensemble techniques

Ensemble techniques in ML enhance predictive accuracy by combining the strengths of multiple models. These methods address the limitations and biases of individual models, producing more robust and accurate outcomes by aggregating their predictions. Ensemble methods such as stacking, bagging, voting, and random subspace, combine multiple models to enhance predictive performance. Stacking uses a meta-learner to integrate base

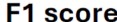 PeerJ Computer Science

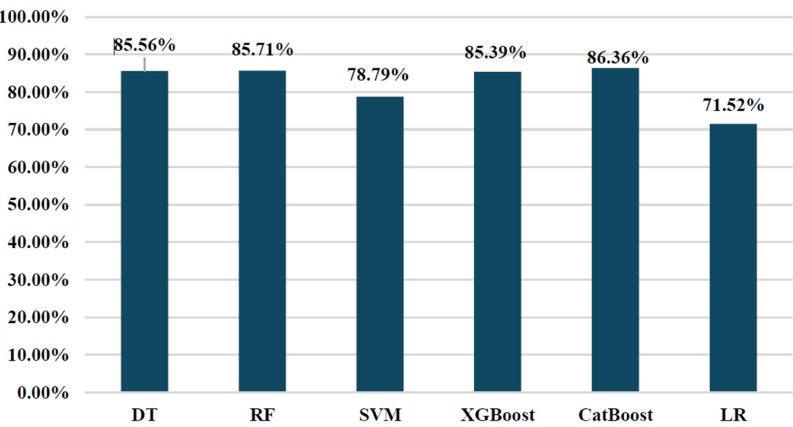

**Figure 12 F1-score for ML techniques.**

model predictions, capturing complex relationships but requiring more computational resources. Bagging trains base learners on random data subsets to reduce variance and overfitting, especially for high-variance models like decision trees. Voting combines model predictions through majority voting or averaging and works well when models are diverse and complementary. The Random Subspace method creates base models by randomly selecting feature subsets, encouraging specialization and diversity in the ensemble, leading to improved overall predictions.

The results revealed that the stacking ensemble outperformed individual base classifiers in terms of overall predictive performance. The stacking ensemble effectively classified both fraudulent and non-fraudulent transactions, as evidenced by its elevated accuracy, precision, recall, and F1-score. The findings provide evidence that stacking is an excellent strategy for ensemble learning, showing the potential of this technique to improve the resilience and reliability of predictive models in applications that are used in the real world. Table 2 shows the results after applying Stacking ensemble method. Figure 13 shows the F1-score for stacking ensemble model.

To validate the observed performance differences between stacking and individual base models, paired t-tests were conducted across the cross-validation folds. Results showed that stacking significantly outperformed SVM and Logistic Regression in terms of F1-score ($p < 0.05$), with 95% confidence intervals (CIs) for the F1 difference being entirely positive. In contrast, stacking performed slightly worse than random forest and CatBoost, as indicated by negative CIs and significant $p$-values. No statistically significant differences were found between stacking and Decision Tree or XGBoost ($p > 0.05$). These findings suggest that while stacking generally improves predictive performance over weaker models, it is not universally superior across all classifiers. As a future extension, validating stacking on multiple datasets would provide further generalizability to the results.

The results presented in the table offer a nuanced comparison between the stacking hybrid ML model and other individual ML algorithms in the context of credit card transaction cyber fraud detection, with a particular emphasis on the F1-score. Comparing

**Table 2 Results after applying stacking ensemble.**

| ML | Accuracy (%) | Precision (%) | Recall (%) | F1-score (%) | AUC (%) | *p*-value (*vs* Stacking) | 95% CI of F1 difference |
|---|---|---|---|---|---|---|---|
| DT | 99.93 | 89.89 | 81.63 | 85.56 | 90.80 | 0.2027 | [−0.0161 to 0.0550] |
| RF | 99.96 | 97.40 | 76.53 | 85.71 | 97.25 | 0.0199 | [−0.0450 to −0.0067] |
| SVM | 99.94 | 97.02 | 66.33 | 78.79 | 95.13 | 0.0225 | [0.0083–0.0635] |
| XGBoost | 99.95 | 95.00 | 77.55 | 85.39 | 97.83 | 0.0628 | [−0.0261 to 0.0011] |
| CatBoost | 99.96 | 97.44 | 77.55 | 86.36 | 98.37 | 0.0315 | [−0.0423 to −0.0033] |
| LR | 99.92 | 88.06 | 60.20 | 71.52 | 97.01 | 0.0047 | [0.0593–0.1727] |
| Stacking hybrid | 99.96 | 98.73 | 79.59 | 88.14 | 89.80 | —— | —— |
| Voting | 99.95 | 97.37 | 75.51 | 85.06 | 97.45 | —— | —— |
| Subspace Random | 99.95 | 95.95 | 72.45 | 82.56 | 86.22 | —— | —— |
| Bagging | 99.95 | 97.30 | 73.47 | 83.72 | 86.73 | —— | —— |

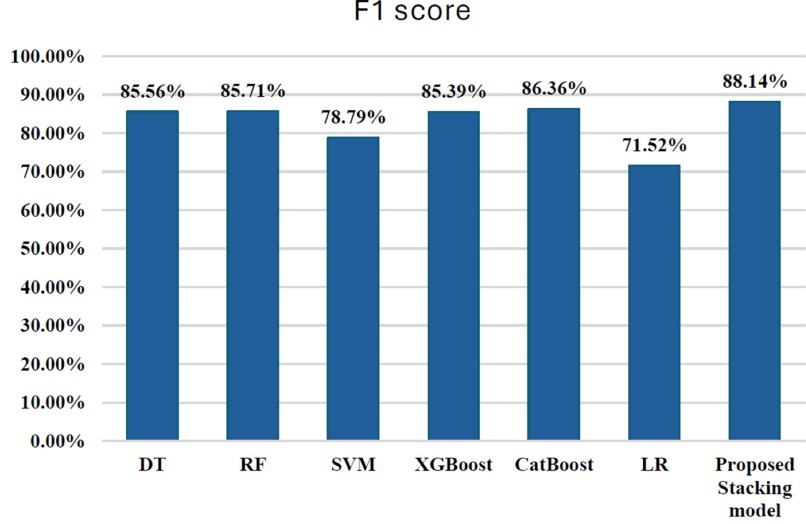

**Figure 13 F1-score for stacking ensemble model.**

the F1-scores of the stacking hybrid ML with those of individual algorithms reveals a notable disparity. The Stacking ensemble model exhibits a significantly higher F1-score compared to all individual algorithms, underscoring the effectiveness of the ensemble approach in bolstering predictive accuracy and robustness. For instance, the Stacking ensemble model achieved an F1-score of 88.14%, surpassing the F1-scores of individual algorithms such as DT (85.56%), RF (85.71%), SVM (78.79%), LR (71.52%), XGBoost (85.39%), and CatBoost (86.36%). This substantial difference highlights the superiority of the Stacking ensemble model in achieving a balanced trade-off between precision and recall, thereby enhancing its capability in accurately detecting fraudulent transactions. Although ensemble stacking typically enhances predictive performance, in this present research work, the stacking model showed a slightly lower AUC-ROC compared to some individual models. This may be due to meta-model overfitting or complex feature

interactions among base classifiers, particularly under severe class imbalance. Future work could explore simpler meta-models or stronger regularization to improve generalization.

In addition to the promising performance of the stacking model, the success of stacking is largely attributed to the diversity of the base models used. Each base classifier—ranging from decision trees to more complex algorithms like CatBoost and XGBoost—captures different aspects of the data. The meta-learner, which aggregates these individual predictions, is able to synthesize a more generalized and robust model that mitigates the biases or weaknesses of any single classifier. This synergy enhances the stacking model's ability to detect fraudulent transactions with high accuracy, while maintaining a well-balanced trade-off between precision and recall.

To evaluate the computational feasibility of the proposed stacking ensemble model, we compared the training times of individual base classifiers with the full ensemble. Simpler models such as logistic regression (6.4 s) and decision tree (44 s) demonstrated rapid training times. In contrast, more complex classifiers like random forest (25 min 48 s) and support vector machine (14 min 31 s) required substantially longer. The full stacking ensemble, incorporating six base models (DT, RF, SVM, XGBoost, CatBoost, LR) and a meta-classifier, required approximately 1 h and 33 s for training.

Despite the increased training time, the ensemble model achieved superior classification performance, with an F1-score of 0.88 on the minority (fraud) class—outperforming all individual models in terms of recall and overall balance. Since model training is an offline process, the computational cost does not affect real-time deployment. Furthermore, once trained, the model performs predictions efficiently, making it suitable for deployment in large-scale, high-transaction environments such as financial fraud detection systems. These findings support the trade-off between training complexity and improved model generalization in high-stakes, data-intensive applications.

However, while stacking enhances performance, it also introduces a computational overhead due to the need for training multiple models. This can lead to increased training time and resource consumption, particularly when working with large datasets. This trade-off should be considered, particularly when real-time fraud detection is required, as deployment of such models in production may need more computational resources. Future studies may focus on optimizing the meta-model selection or employing faster base models to reduce these computational costs without compromising performance.

The choice of stacking as the ensemble technique for the novel hybrid model stems from its inherent advantages over other ensemble methods. Stacking allows for the combination of diverse base classifiers, each capturing unique aspects of the data, leading to a more comprehensive understanding of the underlying patterns. By leveraging the collective intelligence of multiple models, Stacking synthesizes a robust predictive model that is less prone to individual model biases and overfitting. Moreover, Stacking fosters a collaborative synergy among constituent algorithms, enabling them to complement each other's strengths and mitigate weaknesses.

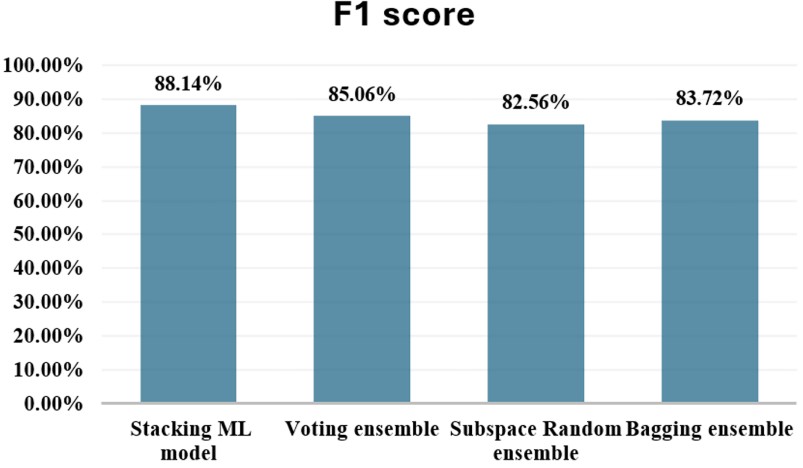

**Figure 14 Comparison with other ensemble techniques.**

## Comparison with other ensemble techniques

In addition to the stacking hybrid ML model, this present research work delves into the exploration of several other ensemble techniques, aiming to thoroughly evaluate their comparative performance. These alternative methodologies comprise a diverse array, including the application of Voting, Subspace Random ensemble, and Bagging ensemble approaches. The empirical findings derived from these experiments are systematically documented in Table 2. Figure 14 shows comparison with other ensemble techniques.

When comparing the ensemble techniques based on F1-score, the stacking hybrid ML emerges as the top performer. With an F1-score of 88.14%, the Stacking model achieves the highest balance between minimizing false positives and false negatives among all the techniques evaluated. This indicates its effectiveness in accurately classifying both fraudulent and non-fraudulent transactions. Following the Stacking model, the Voting ensemble demonstrates a respectable F1-score of 85.06%. The Subspace Random ensemble and Bagging ensemble techniques exhibit F1-scores of 82.56% and 83.72%, respectively. While these scores indicate reasonable performance, they are notably lower compared to both the stacking model and the Voting ensemble. In summary, based on F1-score comparison, The stacking hybrid ML model is the most effective technique for detecting credit card cyber fraud among the ensemble methods that have been evaluated.

The proposed stacking hybrid ML in this present research work advances the state-of-the-art in credit card fraud detection by outperforming prior models in both accuracy and F1-score. Leveraging a diverse ensemble of algorithms—CatBoost, XGBoost, DT, SVM, LR, and RF—this model achieves a high F1-score of 88.14%, surpassing notable benchmarks in the literature. For instance, while models like those by *Hema & Muttipati (2020)* with RF and CatBoost achieved accuracy as high as 99.5% the stacking model here goes further by balancing both false positives and negatives in a way that exceeds conventional ensemble methods. This model's precision is especially pronounced when contrasted with single and ensemble methods reviewed in the literature, such as *Faraj,*

*Mahmud & Rashid (2021)*, where XGBoost led with a maximum score of 0.78%. The stacking model's superior F1-score highlights its robust ability to detect both fraudulent and legitimate transactions with minimal misclassification, marking it as a more effective option for fraud detection.

## CONCLUSION

This research work introduces a novel stacking-based hybrid ML framework specifically designed for cyber fraud detection in credit card transactions. By integrating a diverse ensemble of base classifiers including RF, SVM, LR, DT, XGBoost, and CatBoost, the model harnesses algorithmic diversity to enhance detection robustness. Comprehensive attention was given to data preprocessing, feature engineering, and rigorous model evaluation. Among individual models, CatBoost demonstrated superior performance with an F1-score of 86.36%, effectively minimizing both false positives and false negatives.

The proposed stacking model, which combines base model predictions *via* a meta-classifier, achieved the highest F1-score of 88.14%, outperforming not only the standalone classifiers but also alternative ensemble techniques such as Voting (85.06%), Subspace Random (82.56%), and Bagging (83.72%). These findings underscore the superior discriminative power and generalization capacity of stacking in accurately classifying both fraudulent and legitimate transactions. The results further reinforce the importance of using F1-score as a primary metric when dealing with skewed datasets, where precision–recall trade-offs must be carefully balanced.

Despite its promising results, the study recognizes limitations stemming from the use of a single public dataset, which may introduce biases due to anonymization, class imbalance, or limited diversity in transaction behaviors. These factors may constrain the model's ability to generalize across different financial ecosystems. Future research should therefore include cross-validation on multiple heterogeneous datasets. Additionally, while this work focused on classical ML models, future studies may benefit from incorporating advanced deep learning architectures such as convolutional neural networks (CNNs) and long short-term memory (LSTM) networks to capture temporal and high-dimensional fraud patterns. Finally, although developed for credit card fraud detection, the proposed stacking hybrid framework offers broad applicability across domains such as anomaly detection, cybersecurity, and healthcare diagnostics, demonstrating its versatility as a powerful tool for complex classification tasks.

### Funding
The authors received no funding for this work.

### Competing Interests
The authors declare that they have no competing interests.

## Author Contributions

- Eyad Abdel Latif Marazqah Btoush conceived and designed the experiments, performed the experiments, analyzed the data, performed the computation work, prepared figures and/or tables, authored or reviewed drafts of the article, and approved the final draft.
- Xujuan Zhou conceived and designed the experiments, performed the experiments, authored or reviewed drafts of the article, and approved the final draft.
- Raj Gururajan analyzed the data, authored or reviewed drafts of the article, and approved the final draft.
- Ka Ching Chan performed the computation work, authored or reviewed drafts of the article, and approved the final draft.
- Omar Alsodi analyzed the data, prepared figures and/or tables, and approved the final draft.

## Data Availability

The raw measurements are available in the Supplemental Files.

This dataset is available at Kaggle: https://www.kaggle.com/mlg-ulb/creditcardfraud.

## Supplemental Information

Supplemental information for this article can be found online at http://dx.doi.org/10.7717/peerj-cs.3007#supplemental-information.

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
