# Peer review of "Enhancing credit card fraud detection with a stacking-based hybrid machine learning approach"

_PeerJ Computer Science, doi:10.7717/peerj-cs.3007_

## Round 0.1 · original submission · Major Revisions

This manuscript needs major revision. I was leaning toward a Reject decision, but feel that the topic and coverage have promise. I recommend talking on board Reviewer 1 and Reviewer 3's recommendations. While Reviewer 4 is more positive, on reading the manuscript, and their review, I think their comments are more focused on the area of interest.

In sum, please address the areas for improvement and resubmit with a clear and detailed list of amendments that address the reviewers' "asks".

**Language Note:** The review process has identified that the English language must be improved. PeerJ can provide language editing services - please contact us at [email protected] for pricing (be sure to provide your manuscript number and title). Alternatively, you should make your own arrangements to improve the language quality and provide details in your response letter. – PeerJ Staff

Reviewer 2 ·

Basic reporting

Revise for concise, professional English. For example, rephrase "This method integrates various machine learning models by stacking to improve predictive performance" (Abstract, line 23) to "This method uses stacking to integrate diverse ML models, enhancing predictive performance." Consider professional editing to eliminate errors and enhance readability.
Expand the literature review to critically compare prior stacking implementations and justify this study’s approach. Replace or supplement web-based citations with peer-reviewed sources.
In the introduction, briefly explain why stacking was preferred over alternatives, linking to its theoretical advantages (e.g., meta-model synthesis) and referencing supporting theory or prior findings.

Experimental design

Specify the meta-model and its hyperparameters, and clarify the stacking process (e.g., how base model predictions are fed to the meta-model). Provide a pseudocode snippet or reference a reproducible script.
List exact library versions (e.g., Scikit-learn 1.x.x, CatBoost 1.x.x) and clarify PCA application (own implementation or dataset default). Deposit code and configuration files in a public repository (e.g., GitHub) and link it in the manuscript.
Discuss why 17 features were chosen (e.g., based on explained variance, feature importance scores) and include a brief ablation study or comparison (e.g., 10 vs. 17 vs. 30 features) to validate the choice.
Describe the tuning process (e.g., "Hyperparameters were optimized via grid search over X ranges") to enhance transparency.

Validity of the findings

Investigate and explain the lower AUC-ROC for stacking (e.g., meta-model overfitting, feature interaction effects) in the discussion, or correct if it’s a reporting error.
Include significance tests (e.g., paired t-tests across folds) to validate performance differences, reporting p-values or confidence intervals in Tables 1–3. Qualify claims (e.g., "On this dataset, stacking outperformed…") to avoid implying universal superiority, and suggest multi-dataset validation as a future step.
Expand the discussion to analyze why stacking succeeds (e.g., linking to base model diversity or feature selection) and quantify trade-offs (e.g., training time), tying results back to the introduction’s motivation.
Report per-class metrics (e.g., recall for fraudulent transactions) and consider a stratified evaluation or ROC curve focused on the minority class to better validate fraud detection performance.

Additional comments

The article is suitable for scholarly literature with revisions, as it demonstrates a competent application of ML to a relevant problem and provides reproducible elements. However, it falls short of rigorous scientific standards in several areas:
Language and referencing need refinement for clarity and credibility.
Incomplete replication details and preprocessing justification hinder rigor.
Unaddressed anomalies, lack of statistical validation, and shallow analysis weaken conclusions.

Reviewer 3 ·

Basic reporting

1. What is the unique feature of your method that sets it out from the other stacking-based fraud detection system presented in the literature? Which one is the base learners, pre-processing, meta-learner or a combination of all?

2. It is noteworthy that the proposed method offers a solution that goes beyond implementing stacking just for fraud detection;

3. It is also an improvement to be made to the related work section where the authors could have offered a critical analysis of the existing work done on stacking and ensemble in the context of fraud detection. It will also be useful if you could elaborate on the limitations of prior work and explain how your approach avoids them.

4. The literature review should be expanded in order to compare your work to state of the art research which has been published in Q1 journals and conferences on both fraud detection and ensemble methods.

5. Explain the theoretical basis for your own particular stacking arrangement and why it should be particularly effective for this problem?

6. You are required to provide an elaboration of the time complexity that your stacking ensemble requires to learn, as well as a comparison with the single base classifiers. How feasible is your strategy for large-scale, high-transaction environment?

7. Although you have mentioned the constraint of having only one dataset in the conclusion, could you explain possible bias in the current dataset and possible effects of the biases on the generalization of the results? What else could be done in the future to validate the model on other datasets?

8. It would have been possible to enhance Figure 5 for the purpose of readability, for instance, by making bar labels clearer and using better graphical design.

9. When referring to the confusion matrices in the Figures 6-11, it would be interesting to know if there are more options for visualizations of the confusion matrices such as heat maps or normalized matrices to emphasize certain aspects in the matrices.

10. It would be helpful if Tables 2 and 3 could be combined or restructured in a more concise manner.

Experimental design

N/A

Validity of the findings

N/A

Additional comments

N/A

Reviewer 4 ·

Basic reporting

• The manuscript is well-structured and follows the standard format for academic research papers.
• The language is professional, clear, and concise. However, some sections contain minor syntax errors that could be improved for better readability. E.g across the document, “This study” , “this research” is being used, instead of using so authors are suggested to use “The present research program” or “proposed research work” or “present research work” as appropriate. Correct the formatting issue for section 1.6 heading.
• The introduction adequately introduces the subject, highlighting the increasing complexity of cyber fraud and the limitations of conventional fraud detection techniques.
• The literature review is comprehensive and provides a strong background on existing methodologies, effectively positioning the proposed approach within the research landscape. Authors are advised to cite the relevant research works for all the techniques in discussed in the article, DT, RF, SVM, voting, subspace etc… cite the relevant base papers for those techniques.

Authors may draw insights from https://doi.org/10.1080/23080477.2020.1783491 and cite the paper appropriately.

• In Related work section, correct the first sentence by removing “in datasets” word
• Figures and tables are relevant, well-labeled, and support the research findings. However, ensuring consistency in formatting would enhance readability.

Experimental design

• The study falls within the journal's scope and aligns with contemporary research in AI applications for fraud detection.
• The methodology is clearly described, with a rigorous experimental setup that includes detailed explanations of the machine learning techniques used.
• The dataset selection (European credit card transaction dataset from Kaggle) is appropriate, and preprocessing steps such as feature selection and PCA are well-explained.
• “Feature Scaling” section mentioning that the work has used “Robust Scalar” but in the it is observed that “Standard Scaling” is being used. Furthermore, the researchers have not developed any scaling techniques, they have utilized the exiting scaling techniques, so the section should be updated accordingly.
• The use of multiple base models (DT, RF, SVM, XGBoost, CatBoost, and LR) and a stacking ensemble meta-model is justified.
• The evaluation criteria, including F1-score, precision, recall, and accuracy, are relevant and well-documented.
• Ethical considerations, such as data privacy and fairness, are not explicitly discussed and could be elaborated on.

Validity of the findings

• The experimental results demonstrate the effectiveness of the proposed stacking ensemble model.
• The results are statistically significant, with performance metrics showing an improvement over individual models and other ensemble techniques. However, the work is compared against the base models only.
• The discussion section effectively interprets the findings, linking them back to research objectives and literature.
• The study identifies limitations, such as dataset dependency and the need for further exploration with deep learning models.
• Future research directions, including testing on multiple datasets and integrating deep learning techniques, are well-articulated.

Additional comments

The model is compared with based models only. Authors may consider to extend the work by using explainable AI concepts (may be in future).

---

## Round 0.2 · accepted · Accept

All required changes have been made to a satisfactory degree.

Reviewer 2 ·

Basic reporting

The main comments raised in the first round have been addressed and the revised version can be recommended for acceptance.

Experimental design

The main comments on this section raised in the first round have been addressed.

Validity of the findings

The main comments on this section raised in the first round have been addressed.

Reviewer 3 ·

Basic reporting

The authors did all required revisions. I have no new comments.

Experimental design

N/A

Validity of the findings

N/A

Additional comments

N/A

Reviewer 4 ·

Basic reporting

Authors have improved the revised manuscript, and satisfactorily addressed my earlier expressed concerns.

Experimental design

Addressed in the revised manuscript.

Validity of the findings

Addressed in the revised manuscript.